# Role of the upper airway microbiota in respiratory virus and bacterial pathobiont dynamics in the first year of life

Matthew S. Kelly [1,2,3] ✉, Pixu Shi[4], Sifelane C. Boiditswe[1], Emily Qin[4], Andrew P. Steenhoff [1,5,6], Tiny Mazhani[7], Mohamed Z. Patel[7], Coleen K. Cunningham [8], John F. Rawls [3], Kathy Luinstra[9], Jodi Gilchrist[9], Julia Maciejewski[9], Jillian H. Hurst [2], Patrick C. Seed[10], David Bulir[11] & Marek Smieja[9,12]

The mechanisms by which respiratory viruses predispose to secondary bacterial infections remain poorly characterized. Using 2,409 nasopharyngeal swabs from 300 infants enrolled in a prospective cohort study in Botswana, we perform a detailed analysis of factors that influence the dynamics of bacterial pathobiont colonization during infancy. We quantify the extent to which viruses increase the acquisition of *Haemophilus influenzae*, *Moraxella catarrhalis*, and *Streptococcus pneumoniae*. We provide evidence of cooperative interactions between these pathobionts while identifying host characteristics and environmental exposures that influence the odds of pathobiont colonization during early life. Using 16S rRNA gene sequencing, we demonstrate that respiratory viruses result in losses of putatively beneficial *Corynebacterium* and *Streptococcus* species that are associated with a lower odds of pathobiont acquisition. These findings provide important insights into viral-bacterial relationships in the upper respiratory tract of direct relevance to respiratory infections and suggest that the bacterial microbiota is a potentially modifiable mechanism by which viruses promote bacterial respiratory infections.

Acute respiratory infections (ARIs) are the most common infections across the lifespan and are associated with substantial morbidity and mortality, particularly during childhood. ARIs accounted for 2.2 million deaths in 2021, including >500,000 deaths among children <5 years of age[1]. Although respiratory viruses are detected in the majority of ARIs among children, most severe or fatal infections are caused by bacteria, including *Streptococcus pneumoniae* (pneumococcus), *Haemophilus*

*influenzae*, *Moraxella catarrhalis*, and *Staphylococcus aureus*[2–4]. These bacteria are classified as "pathobionts" because they frequently colonize the nasal passages without causing clinical illness but can cause infection when the local microenvironment is altered, as may occur with respiratory virus infection. Recognition of the relationship between respiratory virus infections and ARIs caused by these bacterial pathobionts dates back to the 1918–1919 influenza pandemic, during which

[1]Botswana-University of Pennsylvania Partnership, Gaborone, Botswana. [2]Division of Pediatric Infectious Diseases, Duke University, Durham, NC, USA. [3]Department of Molecular Genetics and Microbiology, Duke University, Durham, NC, USA. [4]Department of Biostatistics and Bioinformatics, Duke University, Durham, NC, USA. [5]Global Health Center, Children's Hospital of Philadelphia, Philadelphia, PA, USA. [6]Division of Pediatric Infectious Diseases, Children's Hospital of Philadelphia, Philadelphia, PA, USA. [7]University of Botswana School of Medicine, Gaborone, Botswana. [8]Division of Pediatric Infectious Diseases, University of California, Irvine, Orange, CA, USA. [9]Infectious Disease Research Group, Research Institute of St. Joe's Hamilton, Hamilton, ON, Canada. [10]Department of Pediatrics, Feinberg School of Medicine, Northwestern University, Chicago, IL, USA. [11]Department of Chemical Engineering, McMaster University, Hamilton, ON, Canada. [12]Department of Pathology and Molecular Medicine, McMaster University, Hamilton, ON, Canada. ✉e-mail: mkelly3@uams.edu

the majority of influenza-associated deaths were attributable to secondary bacterial pneumonia[5,6]. Recent clinical studies have reported associations between respiratory virus infections and bacterial pneumonia caused by *S. aureus*[7,8] and *S. pneumoniae*[9,10], while respiratory viruses have been shown in animal models to predispose to *H. influenzae* pneumonia[11] and acute otitis media caused by *M. catarrhalis*[12]. Moreover, during the COVID-19 pandemic, substantial reductions in invasive infections caused by *H. influenzae*[13] and *S. pneumoniae*[13–15] correlated with the reduced circulation of respiratory viruses other than severe acute respiratory virus coronavirus 2 (SARS-CoV-2), often in the context of stable prevalence of colonization by these pathobionts. However, despite these epidemiological studies, the precise impacts of respiratory virus infections on the colonization dynamics of bacterial respiratory pathobionts remain poorly characterized.

To colonize the upper respiratory tract (URT), bacterial pathobionts must evade myriad host defenses, including mechanical barriers, immune cells, antibodies, and host antimicrobial peptides. Additionally, these bacteria must co-colonize within the microbial communities that inhabit the URT mucosa. This URT microbiota is increasingly recognized to resist pathobiont colonization and invasion through competition for resources, production of antimicrobial proteins and secondary metabolites, and modulation of host immune responses[16–18]. Most prior studies of this resistance to pathobiont colonization in the URT were cross-sectional, used culture-based methods, and focused on relationships between bacterial pathobionts. These studies have generally reported that cocultivation of *H. influenzae*, *M. catarrhalis*, and *S. pneumoniae* is observed more frequently than would be expected based on the prevalence of each species[19–24], while a negative association between URT colonization by *S. aureus* and *S. pneumoniae* has been observed in many[20–23], but not all[24,25], pediatric studies. Few studies investigated interactions between these bacterial pathobionts using a longitudinal study design and molecular detection methods. Moreover, surprisingly little is known regarding the extent to which other bacterial species in the URT microbiota influence colonization by these pathobionts. An improved understanding of these microbial relationships could lead to novel strategies to prevent ARIs by modifying the URT microbiota.

In this study, we investigated relationships between respiratory viruses, the URT bacterial microbiota, and bacterial respiratory pathobionts in 300 infants in Botswana who were followed during the first year of life. We characterize the impact of respiratory virus infections on the acquisition of bacterial pathobionts and identify cooperative relationships between *H. influenzae*, *M. catarrhalis*, and *S. pneumoniae* within the human URT. We demonstrate that the composition of the URT microbiota predicts the acquisition of bacterial pathobionts during infancy and provide evidence that respiratory viruses predispose to pathobiont acquisition, at least in part through modification of the URT microbiota. Finally, we identify potential relationships between bacterial pathobionts and other URT microbes that could be leveraged to develop novel biotherapeutics for ARI prevention.

## Results
### Respiratory virus infections and bacterial pathobiont colonization occur frequently during infancy
As previously described, we collected nasopharyngeal swab samples ($n = 2409$) monthly (infant age 0–6 months) or bimonthly (infant age >6–12 months) from 300 mother-infant dyads recruited at urban and rural public clinics in southern Botswana[16]. We tested infant nasopharyngeal samples for respiratory viruses and common bacterial respiratory pathobionts by PCR. We determined the composition of the bacterial microbiota by sequencing the V4 region of the 16S ribosomal RNA (rRNA) gene (Fig. 1a). Infants were born vaginally, had a median [interquartile range (IQR)] birth weight of 3085 g (2850 g, 3395 g), and were predominantly breastfed (Table 1). We identified one

or more respiratory viruses in 639 of 2372 (27%) tested samples, with rhinovirus/enterovirus (sample prevalence of 21%), adenovirus (3%), and respiratory syncytial virus (RSV; 2%) being the most frequently detected viruses (Supplementary Table 1). The prevalence of respiratory virus detection increased with age during infancy, approaching 40% at 12 months of age (Fig. 1b; logistic regression, $p < 0.0001$). The prevalence of colonization by bacterial respiratory pathobionts also varied during infancy (Fig. 1c). Colonization by *H. influenzae*, *M. catarrhalis*, and *S. pneumoniae* increased progressively during infancy (logistic regression, $p < 0.0001$ for all comparisons). In contrast, the prevalence of *S. aureus* colonization declined with age (logistic regression, $p < 0.0001$) after peaking at 1 month of age.

ARI symptoms, identified by the presence of cough, rhinorrhea, or nasal congestion, were reported at 610 of 2,409 (25%) study visits and were strongly associated with respiratory virus detection [mixed effect logistic regression; odds ratio (OR): 3.61, 95% confidence interval (CI): 2.89–4.51]. However, the prevalence of ARI symptoms varied based on the respiratory virus detected. ARI symptoms were reported at >70% of visits at which influenza viruses, human metapneumovirus, parainfluenza virus type 3, or RSV were detected, but at only 35% and 41% of visits with detection of adenovirus or rhinovirus/enterovirus, respectively (Supplementary Table 1). Acquisition of *H. influenzae* (mixed effect logistic regression; OR: 1.74, 95% CI: 1.31-2.33), *M. catarrhalis* (OR: 1.55, 95% CI: 1.18-2.03), and *S. pneumoniae* (OR: 1.56, 95% CI: 1.18-2.07) were each associated with ARI symptoms after accounting for respiratory virus detection and the acquisition of other pathobionts, as was a higher pneumococcal carriage density among infants newly colonized with *S. pneumoniae* (per $\log_{10}$ increase in copies/mL; OR: 1.19, 95% CI: 1.07-1.32). Infants acquiring *H. influenzae* (20%), *M. catarrhalis* (16%), or *S. pneumoniae* (15%) were only slightly more likely to receive antibiotics than infants who did not acquire one of these pathobionts (14%), suggesting that URT colonization by these bacterial respiratory pathobionts during infancy is often associated with ARI symptoms in the absence of established infection.

### Respiratory viruses and environmental exposures associated with bacterial pathobiont colonization
We then evaluated the impact of coincident respiratory virus infections and other clinical factors on the acquisition of bacterial pathobionts. We found that respiratory virus detection was associated with higher odds of acquiring *H. influenzae* (Fig. 2a, Table 2; mixed effect logistic regression; OR: 1.44, 95% CI: 1.06-1.96), *M. catarrhalis* (OR: 1.33, 95% CI: 0.96–1.87), and *S. pneumoniae* (OR: 1.83, 95% CI 1.33–2.52), and a lower odds of *S. aureus* acquisition (OR: 0.45, 95% CI: 0.29-0.70). The odds of acquiring *H. influenzae*, *M. catarrhalis*, and *S. pneumoniae* increased in association with respiratory virus detections accompanied by ARI symptoms ($n = 287$) but not with asymptomatic respiratory virus detections ($n = 352$) (Supplementary Table 2). The acquisition of these pathobionts tended to increase with infections involving rhinovirus/enterovirus only ($n = 468$) and infections involving other respiratory viruses ($n = 171$), although the odds of acquiring *H. influenzae* and *M. catarrhalis* were higher with infections caused by other respiratory viruses (Supplementary Table 3). Among infants with pneumococcal carriage, respiratory virus detection was associated with an approximately four-fold increase in the *S. pneumoniae* colonization density [Fig. 2b; median (IQR): 7.4 (6.6–7.8) vs. 7.0 (6.0–7.5) $\log_{10}$ copies/mL; linear regression, $p = 0.006$]. The density of pneumococcal colonization tended to be higher among infants with symptomatic compared to asymptomatic respiratory virus detections [median (IQR): 7.5 (6.9–7.9) vs. 7.3 (6.5–7.8) $\log_{10}$ copies/mL; $p = 0.07$]. However, pneumococcal colonization density was similar between infants with rhinovirus/enterovirus only and those with other respiratory virus infections [median (IQR): 7.4 (6.6–7.8) vs. 7.3 (6.6–7.8) $\log_{10}$ copies/mL; $p = 0.73$].

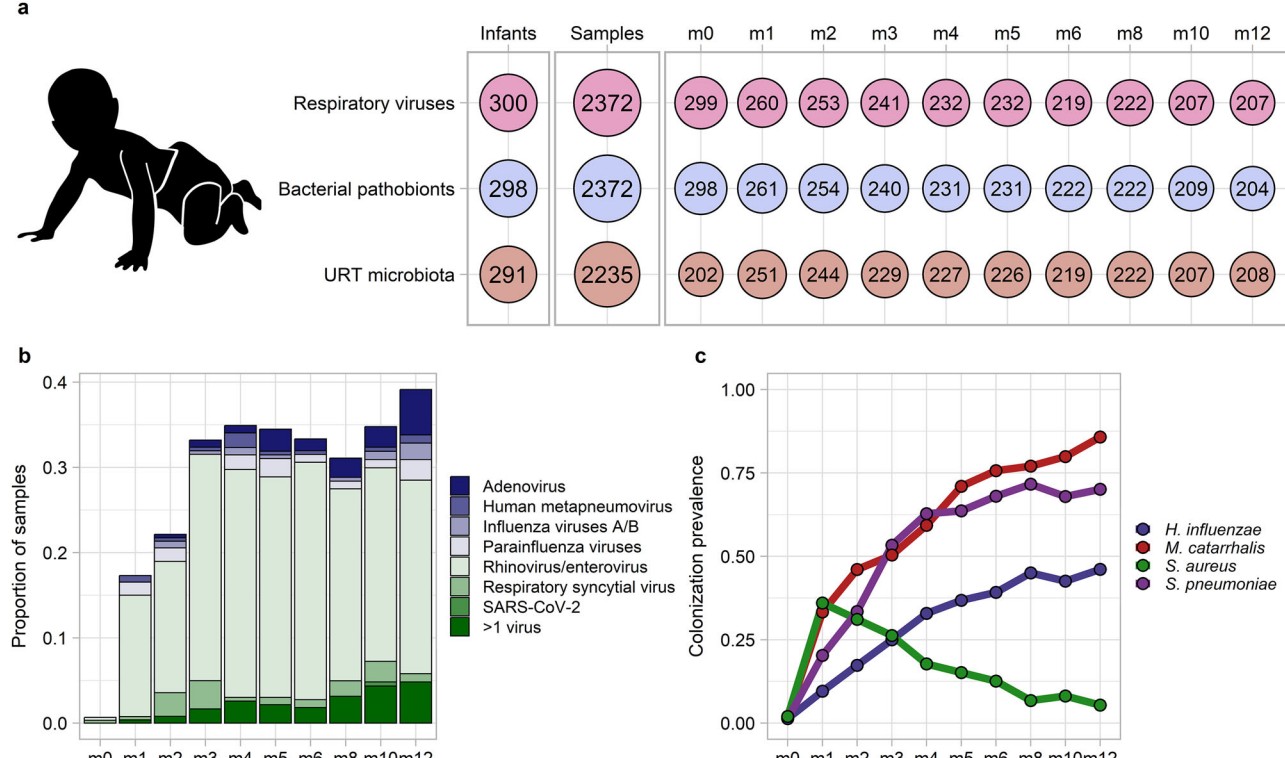

**Fig. 1 | Respiratory viral infections and bacterial pathobiont colonization during infancy. a** Overview of the number of infants and samples from which data were available for respiratory virus testing by PCR, identification of bacterial pathobiont colonization by PCR, and characterization of the upper respiratory tract microbiota through sequencing of the 16S ribosomal RNA gene. The number of samples with available data is further shown by the age of sample collection in

months (m0–m12). "Crawling Baby" icon by Cara Rubley from Noun Project CC BY 3.0. **b** Stacked histogram showing the prevalence of detection of specific respiratory viruses by age of sample collection. **c** Line plots depicting the prevalence of nasopharyngeal colonization by bacterial pathobiots by age of sample collection. Source data are provided as a Source Data file. (URT, upper respiratory tract; SARS-CoV-2, severe acute respiratory syndrome coronavirus 2).

Several infant and sample characteristics were also associated with the odds of acquiring specific bacterial pathobionts (Table 2). Each additional child <5 years of age in the household was associated with higher odds of acquiring *H. influenzae* (mixed effect logistic regression; OR: 1.21, 95% CI: 1.05-1.41), *M. catarrhalis* (OR: 1.14, 95% 0.99-1.32), and *S. pneumoniae* (OR: 1.26, 95% CI: 1.09-1.46). Breast-feeding was associated with lower odds of acquiring *S. aureus* (OR: 0.49, 95% CI: 0.27-0.87), while the odds of *S. aureus* acquisition was higher during the rainy season than during the dry season (OR: 1.58, 95% CI: 1.11-2.26). In contrast, the odds of acquiring *M. catarrhalis* was lower during the rainy season (OR: 0.55, 95% CI: 0.41-0.75), but higher among infants living in households that use solid fuels for cooking or heating (OR: 1.47, 95% CI: 1.02-2.10). The odds of acquiring *S. pneumoniae* were higher in urban settings compared to rural settings (OR: 1.56, 95% CI: 1.09–2.23) but were not significantly associated with the number of 13-valent pneumococcal conjugate vaccine (PCV-13) doses received (OR: 1.15, 95% CI: 0.91–1.44). Among infants with pneumococcal carriage, a higher colonization density was observed with younger age (linear regression; $p = 0.007$), a greater number of children in the household ($p = 0.002$), and receipt of antibiotics since the prior study visit ($p = 0.004$). PCV-13 vaccination status was not significantly associated with the density of pneumococcal colonization [median (IQR) $\log_{10}$ copies/mL; 0 doses: 7.5 (6.7–7.9); 1 dose: 7.3 (6.6–7.6); 2 doses: 7.1 (6.1–7.7); 3 doses: 7.0 (6.0–7.7); $p = 0.14$]. Thus, in addition to confirming the substantial role of respiratory virus infections, our findings suggest that feeding practices, season, location of residence, and exposure to other children and smoke from solid fuels influence the acquisition of bacterial pathobionts during infancy.

## Patterns of pathobiont colonization during infancy support cooperative interspecies relationships

We next sought to evaluate for interactive relationships between bacterial respiratory pathobionts within the infant URT. As reported by previous cross-sectional studies[19–22], we observed positive correlations for the co-detection of *H. influenzae*, *M. catarrhalis*, and *S. pneumoniae* within samples (Fig. 2c; Fisher's exact tests; *H. influenzae*-*M. catarrhalis*, OR: 4.46, 95% CI: 3.60–5.54; *M. catarrhalis*-*S. pneumoniae*, OR: 8.19, 95% CI: 6.77–9.92; *H. influenzae*-*S. pneumoniae*, OR: 5.67, 95% CI: 4.60–7.02), as well as negative correlations for the co-detection of *S. aureus* and other pathobionts (*S. aureus*-*H. influenzae*, OR: 0.44, 95% CI: 0.33-0.59; *S. aureus*-*M. catarrhalis*, OR: 0.33, 95% CI: 0.26-0.42; *S. aureus*-*S. pneumoniae*, OR: 0.46, 95% CI: 0.36-0.58). We then performed longitudinal analyses to evaluate if colonization by each bacterial pathobiont was associated with the subsequent acquisition of other bacterial pathobionts (Fig. 2a, Table 2). We found that *M. catarrhalis* colonization was associated with higher odds of acquiring *H. influenzae* (mixed effect logistic regression; OR: 1.82, 95% CI: 1.28-2.60) and *S. pneumoniae* (OR: 1.58, 95% CI: 1.12-2.24), and *S. pneumoniae* colonization was associated with a higher odds of *H. influenzae* acquisition (OR: 1.42, 95% CI: 1.01-1.98). Additionally, among infants with pneumococcal carriage, concurrent *H. influenzae* colonization was associated with an approximately three-fold higher *S. pneumoniae* colonization density [Fig. 2b; median (IQR): 7.3 (6.5-7.7) vs. 7.0 (5.9-7.6) $\log_{10}$ copies/mL; linear regression; $p < 0.0001$]. The odds of *S. aureus* acquisition tended to be lower with preceding *S. pneumoniae* colonization (mixed effect logistic regression; OR: 0.66, 95% CI: 0.41-1.06), but preceding *S. aureus* colonization was not associated with the acquisition of other pathobionts (Fig. 2a). These findings provide further supportive evidence of potential cooperative

**Table 1 | Characteristics of the 300 infants in the study population**

| Subject Characteristics | N | % |
|---|---|---|
| Infant sex | | |
| Female | 167 | 56% |
| Male | 133 | 44% |
| Median (IQR) infant birth weight, g | 3085 | (2850, 3395) |
| Infant HIV status | | |
| HIV-unexposed, uninfected | 212 | 71% |
| HIV-exposed, uninfected | 87 | 29% |
| HIV-exposed, infected | 1 | <1% |
| Location of residence | | |
| Rural | 115 | 38% |
| Urban | 185 | 62% |
| Electricity in the home | 210 | 70% |
| Household use of solid fuels | 164 | 55% |
| Maternal educational level | | |
| None or primary | 21 | 7% |
| Secondary | 238 | 79% |
| Tertiary | 41 | 14% |
| Median (IQR) number of household members | | |
| <5 years | 0 | (0, 1) |
| 5–17 years | 1 | (0, 3) |
| ≥18 years | 3 | (2, 4) |
| Enrollment season | | |
| Dry (April to October) | 190 | 63% |
| Rainy (November to March) | 110 | 37% |
| One or more PCV-13 doses during follow-up | 248 | 83% |
| Infant breastfeeding, any | 250 | 83% |
| Infant antibiotic exposure, any | | |
| Amoxicillin | 118 | 39% |
| Metronidazole | 31 | 10% |
| Trimethoprim-sulfamethoxazole | 75 | 25% |

IQR, interquartile range; PCV-13, 13-valent pneumococcal conjugate vaccine.

interactions between *H. influenzae*, *M. catarrhalis*, and *S. pneumoniae* within the human URT. Moreover, the absence of associations between *S. aureus* and other bacterial pathobionts in longitudinal analyses suggests that the negative co-occurrence patterns observed between *S. aureus* and these species in cross-sectional analyses of individual samples are unlikely to result from direct antagonistic interspecies interactions.

**Respiratory virus infections trigger shifts in the URT microbiota that promote pathobiont acquisition**

Given the observed associations between respiratory virus infections and the acquisition of bacterial respiratory pathobionts during infancy, we sought to evaluate the extent to which alterations of the URT bacterial microbiota may contribute to these viral-bacterial relationships. To investigate shifts in overall URT microbiota community structure, we analyzed 16S rRNA gene sequencing data and applied k-medoids clustering on Bray-Curtis distances to classify infant nasopharyngeal samples into eight distinct microbiota types (Fig. 3a,b). URT microbiota types of similar composition were identified using k-means clustering based on Euclidean distances (Supplementary Fig. 1), suggesting that the types identified by k-medoids clustering are robust to both distance metric and clustering algorithm. Excluding samples collected during the birth visit and with subject included as a random effect, the prevalence of microbiota

types in which the most abundant bacterial genera were *Staphylococcus* (mixed effect logistic regression; STA, $p < 0.0001$) and *Corynebacterium* (COR, $p < 0.0001$) declined with age, while microbiota types in which *Haemophilus* (HAE, $p < 0.0001$), *Moraxella* (MOR, $p < 0.0001$), or *Corynebacterium*, *Dolosigranulum*, and *Moraxella* (CDM, $p < 0.0001$) were most abundant increased during infancy (Fig. 3c). Compared to intervals when no respiratory virus was identified, intervals with respiratory virus infections were more likely to be associated with a shift in URT microbiota type (mixed effect logistic regression, $p = 0.046$). Specifically, respiratory virus infections were associated with a declining prevalence of the *Staphylococcus* (STA, $p = 0.0002$), *Corynebacterium* (COR, $p = 0.03$), and *Corynebacterium/Dolosigranulum* (CD, $p = 0.007$) microbiota types, while the prevalence of the *Haemophilus* (HAE, $p < 0.0001$), *Moraxella* (MOR, $p = 0.10$), and *Streptococcus* (STR, $p < 0.0001$) microbiota types increased (Fig. 3d). These transitions in URT microbiota community states correlated with the subsequent odds of acquiring bacterial pathobionts. Compared to a *Corynebacterium* (COR) microbiota type, infants with a *Haemophilus* (HAE) microbiota type had higher odds of acquiring *M. catarrhalis* (mixed effect logistic regression; OR: 2.89, 95% CI: 1.29-6.45) and *S. pneumoniae* (OR: 3.41, 95% CI: 1.45-8.00). Similarly, a *Moraxella* (MOR) microbiota type was associated with higher odds of acquiring *H. influenzae* (OR: 2.58, 95% CI: 1.40-4.76) and *S. pneumoniae* (OR: 3.10, 95% CI: 1.63-5.87), while a *Streptococcus* (STR) microbiota type was associated with *H. influenzae* acquisition (OR: 2.26, 95% CI: 1.19-4.28). The acquisition of bacterial pathobionts was associated with shifts in URT microbiota composition toward microbiota types in which the most abundant genus contained that pathobiont (Supplementary Fig. 2).

Next, we used MaAsLin2 to fit general linear mixed effect models to evaluate the impact of respiratory virus infection on the relative abundances of specific amplicon sequence variants (ASV) within the infant URT microbiota. For these and other analyses conducted at the ASV level, we aggregated all ASVs that did not have a relative abundance of at least 0.1% in 5% or more of samples into a single "other" category. Respiratory virus infection was associated with decreases in the relative abundances of an ASV classified as *Streptococcus thermophilus/vestibularis/salivarius* (ASV15, $q = 0.18$) and several ASVs classified as *Corynebacterium* species (Fig. 3e), including *C. propinquum/pseudodiphtheriticum* (ASV4, $q = 0.19$; ASV18, $q = 0.15$), *C. accolens/macginleyi* (ASV6, $q = 0.10$), and *C. tuberculostearicum* (ASV86, $q = 0.16$). Most other changes in the relative abundances of specific ASVs associated with respiratory virus infection are consistent with trends in pathobiont colonization that we previously identified by PCR, with increases in the abundances of *Haemophilus* sp. (ASV7, $q < 0.0001$; ASV8, $p = 0.0008$), *M. catarrhalis/nonliquefaciens* (ASV1, $q = 0.10$), and *S. pneumoniae/pseudopneumoniae* (ASV5, $q < 0.0001$), and a decrease in the abundance of *Staphylococcus* sp. (ASV3, $q < 0.0001$) (Fig. 3e). Each of these ASVs was more abundant (Wilcoxon rank-sum tests, $p < 0.0001$ for all comparisons) in children colonized with the pathobiont from that genus than in noncolonized children, indicating that these ASVs are highly likely to contain the pathobiont belonging to that genus (Supplementary Fig. 3).

**The URT microbiota influences the acquisition of bacterial pathobionts during infancy**

Finally, we assessed the extent to which the composition of the URT microbiota influenced the acquisition of pathobionts among infants using both discriminative (random forests) and generative (mixed-effect logistic regression) methods. We first constructed random forest models to determine the utility of the preceding URT microbiota in predicting the acquisition of bacterial respiratory pathobionts at the next study visit. We compared the accuracy of these models to those incorporating infant and sample characteristics such as age and respiratory virus infection. We found that models based on age alone

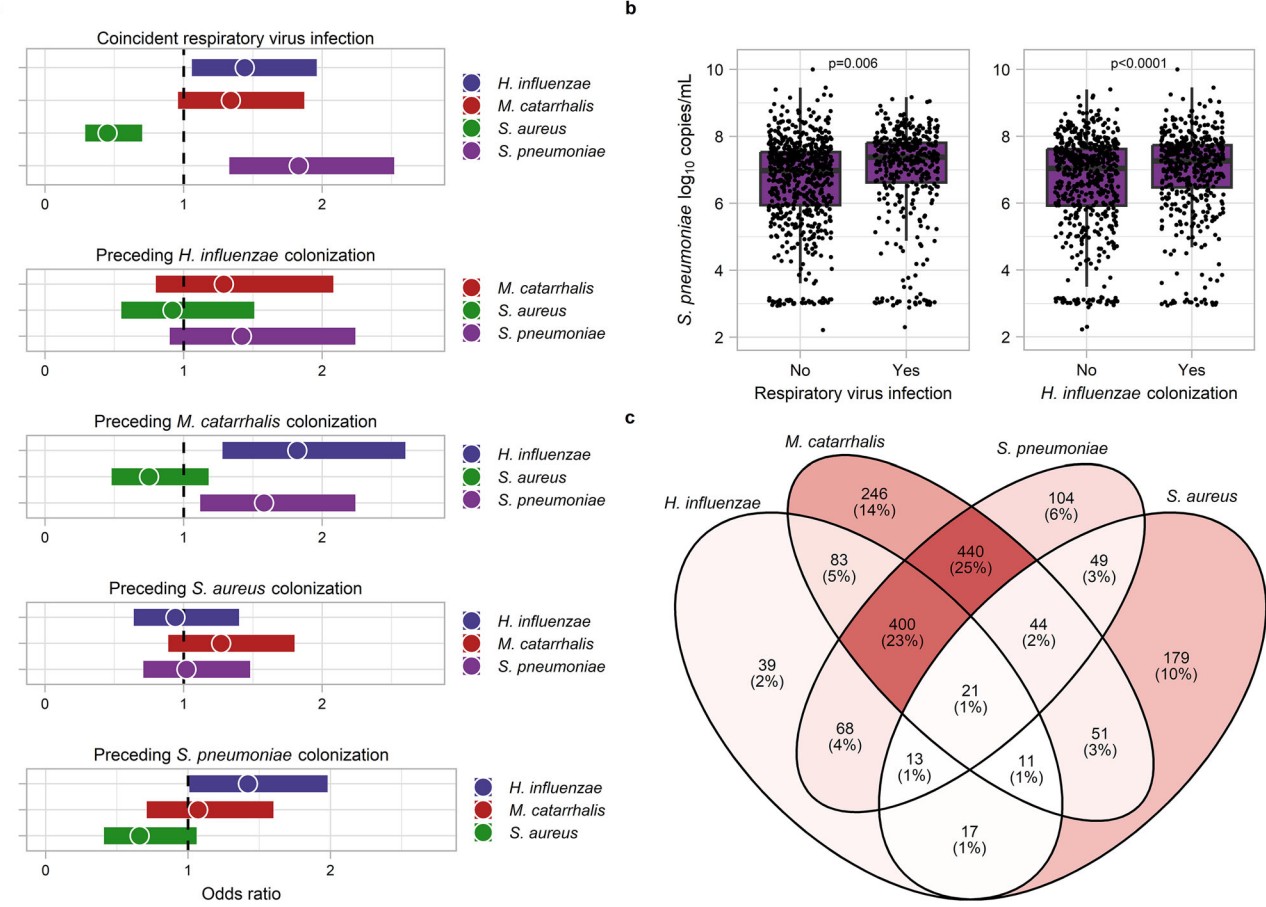

**Fig. 2 | Viral-bacterial and bacterial-bacterial relationships and the dynamics of bacterial pathobiont colonization during infancy. a** Forest plots presenting results from mixed effect logistic regression models evaluating associations between preceding bacterial pathobiont colonization or respiratory virus infection and the subsequent acquisition of specific bacterial pathobionts. **b** Box and whisker plots show the $\log_{10}$ copies/mL of *Streptococcus pneumoniae* DNA detected by quantitative PCR in samples from infants with pneumococcal colonization ($n = 1064$) with and without concurrent respiratory virus detection or *H. influenzae* colonization. The line within each box indicates the median, box edges represent the 25th and 75th percentiles, whiskers extend to 1.5 times the interquartile range,

and individual points represent data from individual samples. *p*-values were estimated using linear regression. The *p*-value for the comparison of *S. pneumoniae* colonization density by concurrent *H. influenzae* colonization status was $1.54 \times 10^{-5}$. **c** Venn diagram depicting the co-occurrence of bacterial respiratory pathobionts within infant upper respiratory samples, including the number (percentage) of samples with detection of specific combinations of these pathobionts. Red shading within the Venn diagram corresponds to the number of samples with each combination of pathobiont colonization. There were 586 (25%) samples from which no bacterial pathobionts were detected (not shown). Source data are provided as a Source Data file.

predicted the acquisition of all bacterial pathobionts better than expected by chance (Fig. 4a). However, the accuracy of these models varied by pathobiont, with the highest accuracy achieved for prediction of *S. aureus* acquisition [area under the receiver operating characteristic curve (AUC-ROC): 0.66, 95% CI: 0.61–0.71]. Adding other infant and sample characteristics improved the predictive accuracy of all models. Models incorporating only the taxonomic composition of the URT microbiota from the preceding sample performed as well as or better than models incorporating infant and sample characteristics for prediction of acquisition of *H. influenzae*, *S. aureus*, and *S. pneumoniae* (Fig. 4a), but not *M. catarrhalis*. The accuracy of these taxonomic models was not substantively improved by the inclusion of infant and sample characteristics or data on the abundances of functional pathways within the URT microbiota as predicted by PICRUSt2[26]. In models incorporating clinical variables and URT microbiota taxonomic composition, microbial taxa were the variables most predictive of pathobiont acquisition in all models, while age and the number of children in the household were the most discriminative clinical variables (Fig. 4b).

We then fit mixed-effect logistic regression models to identify specific URT microbiota features associated with the subsequent

acquisition of bacterial pathobionts. As opposed to random forests, this approach enabled us to account for repeated sampling from the same subject and adjust for a non-linear effect of age at the time of sampling. These analyses identified several microbial taxa associated with pathobiont acquisition (Fig. 4c). First, consistent with analyses based on the detection of pathobionts by PCR, higher relative abundances of *M. catarrhalis/nonliquefaciens* (ASV1) were associated with higher odds of *H. influenzae* ($q = 0.0002$) and *S. pneumoniae* ($q = 0.006$) acquisition. Similarly, *S. pneumoniae/pseudopneumoniae* (ASV5) was positively associated with the acquisition of *H. influenzae* ($q < 0.0001$), while higher abundances of *Haemophilus* sp. (ASV8) were associated with *M. catarrhalis* acquisition ($q = 0.05$). Additionally, we identified several ASVs classified as commensal bacterial species that were negatively associated with bacterial pathobiont acquisition (Fig. 4c), including ASVs that declined in abundance in association with respiratory virus infection. Specifically, higher relative abundances of *C. propinquum/pseudodiphtheriticum* (ASV4) were associated with lower odds of *S. aureus* acquisition ($q = 0.03$), and *C. accolens/macginleyi* (ASV6) was negatively associated with *M. catarrhalis* acquisition ($q = 0.05$). Both *C. tuberculostearicum* (ASV10, $q = 0.02$) and *C. amycolatum* (ASV53, $q = 0.06$) were

**Table 2 | Associations between respiratory virus detections and the odds of acquiring specific bacterial pathobionts during infancy**

| Microbiological testing | H. influenzae acquisition | | M. catarrhalis acquisition | | S. aureus acquisition | | S. pneumoniae acquisition | |
|---|---|---|---|---|---|---|---|---|
| | Odds ratio (95% CI) | p | Odds ratio (95% CI) | p | Odds ratio (95% CI) | p | Odds ratio (95% CI) | p |
| Coincident respiratory virus infection | 1.44 (1.06-1.96) | 0.02 | 1.34 (0.96-1.87) | 0.09 | 0.45 (0.29-0.70) | 0.0004 | 1.83 (1.33-2.52) | 0.0002 |
| Preceding H. influenzae colonization | – | – | 1.29 (0.80-2.08) | 0.30 | 0.92 (0.55-1.51) | 0.72 | 1.42 (0.90-2.24) | 0.13 |
| Preceding M. catarrhalis colonization | 1.82 (1.28-2.60) | 0.0009 | – | – | 0.75 (0.48-1.18) | 0.22 | 1.58 (1.12-2.24) | 0.009 |
| Preceding S. aureus colonization | 0.94 (0.64-1.40) | 0.77 | 1.27 (0.89-1.80) | 0.19 | – | – | 1.02 (0.71-1.48) | 0.91 |
| Preceding S. pneumoniae colonization | 1.42 (1.01-1.98) | 0.04 | 1.07 (0.71-1.60) | 0.75 | 0.66 (0.41-1.06) | 0.09 | – | – |
| **Infant and sample characteristics** | | | | | | | | |
| Male sex | 0.79 (0.55-1.14) | 0.21 | 0.99 (0.70-1.40) | 0.95 | 1.62 (1.05-2.49) | 0.03 | 1.09 (0.77-1.53) | 0.63 |
| Low birth weight (<2500 g) | 1.09 (0.57-2.11) | 0.79 | 0.88 (0.48-1.61) | 0.67 | 0.75 (0.34-1.67) | 0.48 | 1.31 (0.71-2.43) | 0.38 |
| Maternal HIV infection | 1.02 (0.62-1.66) | 0.95 | 0.85 (0.53-1.37) | 0.51 | 0.67 (0.37-1.21) | 0.18 | 1.27 (0.80-2.02) | 0.31 |
| Urban residence | 0.98 (0.67-1.42) | 0.90 | 1.06 (0.74-1.51) | 0.76 | 1.17 (0.75-1.82) | 0.50 | 1.56 (1.09-2.23) | 0.01 |
| Household use of solid fuels | 1.06 (0.72-1.55) | 0.77 | 1.47 (1.02-2.10)– | 0.04 | 1.22 (0.77-1.91) | 0.40 | 1.07 (0.75-1.55) | 0.70 |
| Number of children in the home | 1.21 (1.05-1.41) | 0.01 | 1.14 (0.99-1.32) | 0.06 | 0.95 (0.79-1.15) | 0.59 | 1.26 (1.09-1.46) | 0.002 |
| Rainy season | 1.04 (0.77-1.39) | 0.80 | 0.55 (0.41-0.75) | 0.0002 | 1.58 (1.11–2.26) | 0.01 | 0.78 (0.57-1.05) | 0.10 |
| Number of PCV-13 doses received | – | – | – | – | – | – | 1.15 (0.91-1.44) | 0.24 |
| Current breastfeeding | 0.87 (0.55–1.37) | 0.55 | 0.81 (0.51-1.30) | 0.39 | 0.49 (0.27-0.87) | 0.01 | 1.16 (0.73-1.84) | 0.52 |
| Antibiotics since the prior study visit | 1.00 (0.68-1.47) | 0.99 | 1.16 (0.76-1.78) | 0.49 | 0.48 (0.27-0.85) | 0.01 | 0.75 (0.50-1.14) | 0.18 |

Odds ratios, 95% confidence intervals, and two-sided p-values were estimated using mixed-effects logistic regression models, with age included as a fixed effect. No adjustments were made for multiple comparisons. CI, confidence interval; PCV-13, 13-valent pneumococcal conjugate vaccine.

negatively associated with *S. pneumoniae* acquisition. Higher relative abundances of *Streptococcus thermophilus/vestibular/salivarius* (ASV15) were associated with lower odds of *H. influenzae* acquisition ($q = 0.10$), while *Streptococcus mitis/oralis* (ASV9) was negatively associated with the acquisition of *H. influenzae* ($q = 0.03$), *M. catarrhalis* ($q = 0.14$), and *S. pneumoniae* ($q = 0.009$). Finally, higher abundances of *D. pigrum* (ASV2) were associated with lower odds of *S. aureus* acquisition ($q = 0.03$), and higher abundances of *Veillonella massiliensis* (ASV50) were associated with lower odds of *S. pneumoniae* acquisition ($q = 0.02$). Taken together, these data suggest that respiratory viruses promote bacterial pathobiont acquisition through both the cooperative relationships that exist between *H. influenzae*, *M. catarrhalis*, and *S. pneumoniae* and detrimental effects on commensal microbes that contribute to colonization resistance to these respiratory pathobionts.

## Discussion

In this study of 300 infants in Botswana, we describe the impact of host and environmental factors, respiratory virus infection, and the URT microbiota on the dynamics of bacterial respiratory pathobiont colonization during the first year of life. Our findings characterize in detail the impact of respiratory virus infections on the acquisition of bacterial pathobionts and provide further supportive evidence of cooperative relationships between *H. influenzae*, *M. catarrhalis*, and *S. pneumoniae* within the human URT. We also demonstrate that shifts in URT microbiota composition occurring in association with respiratory virus infection promote pathobiont acquisition. Finally, we identify potential interspecies relationships that, if confirmed through future mechanistic studies, could lead to an improved understanding of microbial ecology in the URT with direct relevance for the prevention of ARIs among both children and adults.

The risk of infections caused by bacterial respiratory pathobionts peaks at the extremes of age. The highest incidences of pneumonia[1] and invasive infections caused by *H. influenzae*[27] and *S. pneumoniae*[28]

occur among young children and older adults, while most cases of acute otitis media occur during early childhood[29,30]. URT colonization by bacterial pathobionts is a necessary prerequisite both for the development of ARIs caused by these pathobionts and for person-to-person transmission[31]. Thus, studies evaluating pathobiont colonization can provide important insights into their population burden and identify modifiable factors that influence the risks of pathobiont acquisition and infection. In this study, we collected serial URT samples from a large cohort of infants in a setting in which risk factors for ARI are highly prevalent[32,33]. We found that respiratory virus infections increased the odds of acquiring *H. influenzae*, *M. catarrhalis*, and *S. pneumoniae* by 44%, 34%, and 83%, respectively. Further, respiratory virus infections were additionally associated with higher colonization densities among infants with *S. pneumoniae* carriage. Children with pneumonia have higher URT pneumococcal colonization densities than healthy children with nasopharyngeal carriage[34,35], suggesting that overgrowth of *S. pneumoniae* in the URT may facilitate invasion of the lower respiratory tract. Moreover, higher URT densities were shown to promote *S. pneumoniae* transmission in a murine model[36]. Notably, we found that respiratory virus detections accompanied by ARI symptoms were associated with increased odds of acquiring *H. influenzae*, *M. catarrhalis*, and *S. pneumoniae*, whereas this association was not observed with asymptomatic respiratory virus detections. Moreover, among infants with *S. pneumoniae* carriage, colonization density tended to be higher in those with symptomatic respiratory virus detections compared to those with asymptomatic respiratory virus detections. These findings are consistent with prior studies suggesting that secondary bacterial infections are more likely to occur following symptomatic respiratory virus infections. For instance, among 367 infants followed during the first year of life, symptomatic respiratory virus infections were associated with an increased risk of acute otitis media, whereas this association was not observed for asymptomatic respiratory virus infections[37]. Symptoms of respiratory

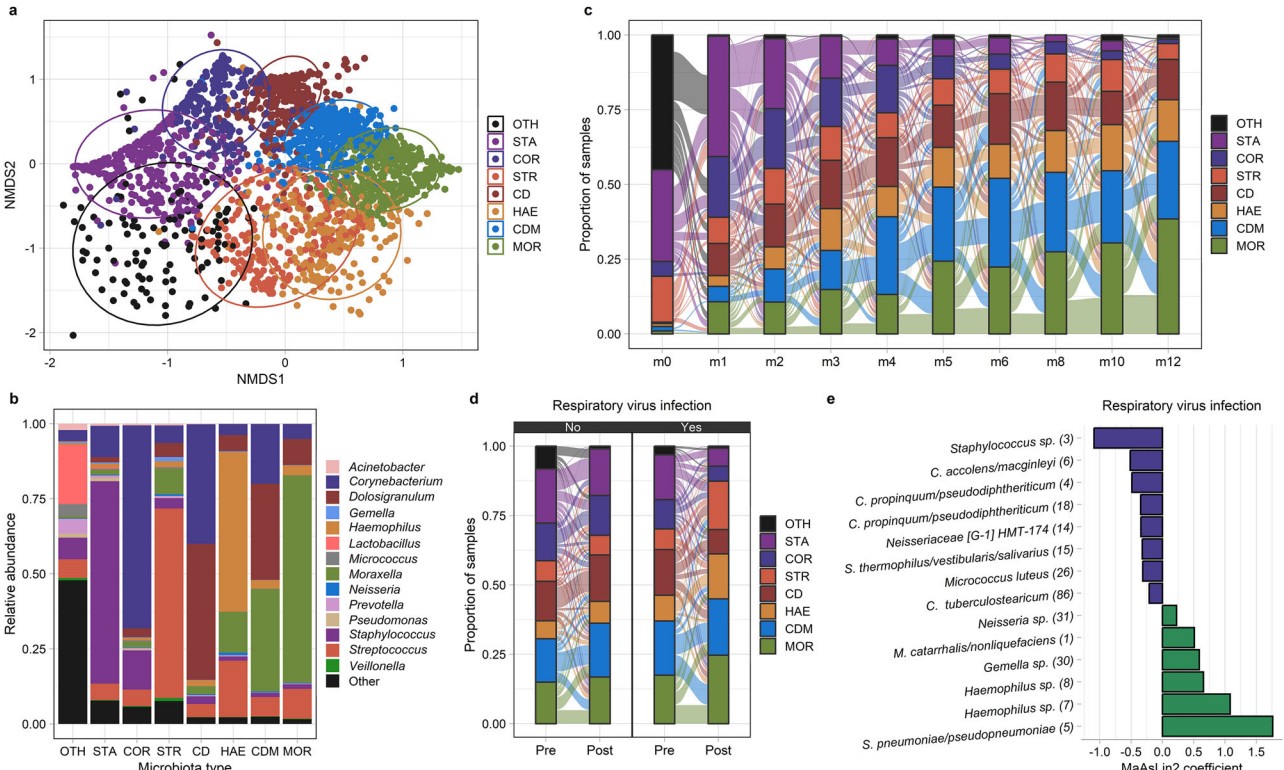

**Fig. 3 | The dynamics of the upper respiratory microbiota among infants in Botswana. a** NMDS plot based on Bray-Curtis distances depicts microbiota composition based on 16S rRNA sequencing of nasopharyngeal swabs collected from infants in Botswana during the first year of life. k-medoids clustering was used to identify 8 distinct microbiota types. Ellipses define the regions containing 80% of all samples that can be drawn from the underlying multivariate t distribution. **b** Relative abundances of highly abundant genera in upper respiratory samples from infants (n = 2,235 samples) by microbiota type. **c, d** Alluvial diagram depicting upper respiratory microbiota community state transitions during the first year of life (**c**) and with respiratory virus infection (**d**). **e** MaAsLin2 was used to fit generalized linear mixed model evaluating the association between respiratory virus infection and the relative abundances of ASVs within the infant upper respiratory microbiota. The coefficients from these models, which correspond to the relative effect sizes of associations, are shown for significant associations (q < 0.20). ASVs that decreased/increased in abundance in association with respiratory virus infection are shown as blue/green bars. The taxonomic classification of each ASV based on BLAST searches is shown followed by a number in parentheses corresponding to the mean relative abundance of this ASV in infant nasopharyngeal samples (ASV1 was the most abundant ASV across all samples). URT microbiota types are named according to the bacterial genus or genera with the highest mean relative abundance: CD, *Corynebacterium/Dolosigranulum*; CDM, *Corynebacterium/Dolosigranulum/Moraxella*; COR, *Corynebacterium*; HAE, *Haemophilus*; MOR, *Moraxella*; STA, *Staphylococcus*; STR, *Streptococcus*; and OTH, "other." Source data are provided as a Source Data file. (ASV, amplicon sequence variant; NMDS, non-metric multidimensional scaling).

virus infections typically result from loss of cellular tight junctions, vascular leakage and edema, increased mucus production, and sloughing of epithelial cells[38–40], all of which can promote bacterial pathobiont adhesion and overgrowth[41–43]. In contrast, we found that respiratory virus infections were actually associated with lower odds of *S. aureus* acquisition. Prior studies suggest that the risk of secondary *S. aureus* pneumonia is likely to vary based on the inciting virus. Influenza viruses[7,8] and SARS-CoV-2[44,45] are the viruses that have been most strongly associated with this complication, while rhinoviruses/enteroviruses accounted for the vast majority of the viral infections identified in our cohort.

Using longitudinal analyses and molecular methods for detecting bacterial pathobionts, we found evidence of cooperative relationships between *H. influenzae*, *M. catarrhalis*, and *S. pneumoniae*. These findings align with those of prior studies, most of which were cross-sectional and used culture-based methods[19–24], and suggest that these pathobionts may act in concert in the human URT to promote colonization and infection. Several potential mechanisms for cooperation between these species have previously been identified. For example, these pathobionts are known to form multi-species biofilms in several chronic respiratory diseases, including in the middle ears of children with chronic otitis media[46,47]. Moreover, the production of virulence factors by these pathobionts may be altered in these biofilms. For example, Cope and colleagues found that *S. pneumoniae* promoted transcription of genes corresponding to type IV pili by *H. influenzae* when these species were co-cultured, while pyruvate oxidase gene expression was upregulated by *S. pneumoniae*[48]. Interestingly, Schaar and colleagues demonstrated that *M. catarrhalis* strains harboring beta-lactamases can release outer membrane vesicles containing these enzymes that prevent antibiotic-induced killing of susceptible strains of *H. influenzae* and *S. pneumoniae*[49]. Consistent with several previous studies of young children[20–23], we identified a negative co-occurrence pattern between *S. aureus* and *S. pneumoniae* in cross-sectional analyses. However, we found no clear evidence of antagonism between these species in longitudinal analyses, suggesting that the negative co-occurrence of *S. aureus* and *S. pneumoniae* reported in studies of young children is unlikely to be explained by direct bacterial interference. Instead, this observation could reflect age-related changes in host mucosal immunity, local micronutrients, or URT microbiota composition, particularly as the antagonism between these species has generally not been observed in older children[50] or adults[21].

We also identified infant characteristics and environmental exposures that were associated with pathobiont colonization during infancy in this setting. First, we found that younger age was associated with a higher pneumococcal colonization density. Prior studies have reported conflicting findings for this association, with higher densities

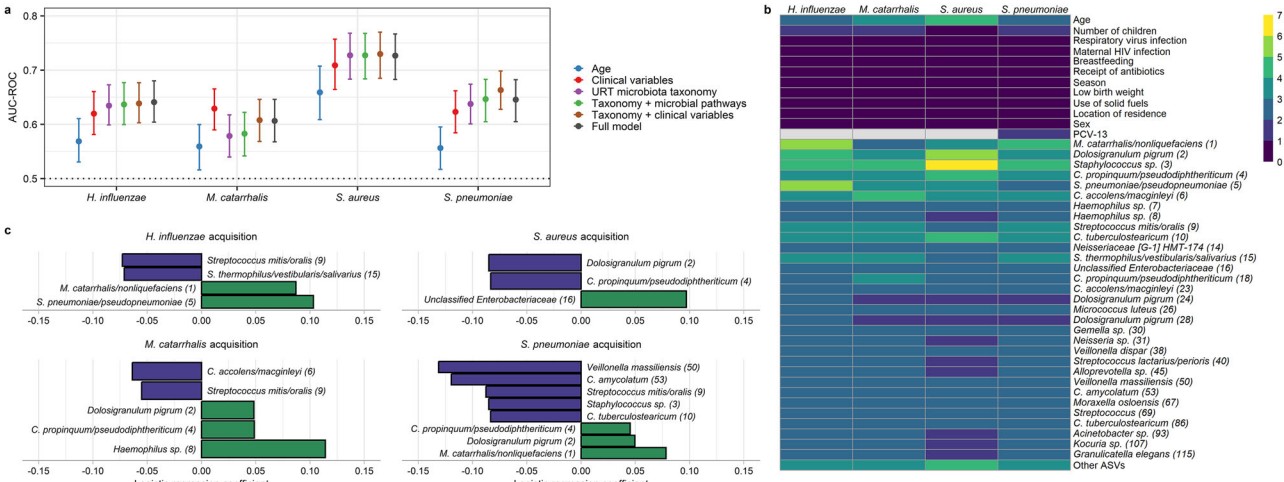

**Fig. 4 | Associations between the upper respiratory microbiota and the acquisition of bacterial pathobionts. a** Forest plots depict the accuracy of random forest models for the prediction of acquisition of bacterial pathobionts. The area under the curve (AUC) is represented by a point, with error bars indicating the 95% confidence interval. Clinical variables included in the models were age, number of children in the household (<5 years of age), respiratory virus infection, maternal HIV infection, breastfeeding, receipt of antibiotics, season, low birth weight, use of solid fuels, location of residency (urban or rural), and 13-valent pneumococcal conjugate vaccine doses (for *S. pneumoniae* only). Data on URT microbiota taxonomy and microbial pathways were from the study visit preceding the visit at which the sample was collected to evaluate for bacterial pathobiont acquisition. For time-varying clinical variables, the data included in the models were collected at the study visit at which the sample was collected to evaluate for pathobiont acquisition. Models predicted pathobiont acquisition at the following number of study visits: *H. influenzae* (*n* = 1319), *M. catarrhalis* (*n* = 827), *S. aureus* (*n* = 1488), and *S.*

*pneumoniae* (*n* = 936). **b** Heatmap depicting variable of importance scores for features in random forest models predicting pathobiont colonization and containing URT microbiota taxonomic data and clinical variables. The taxonomic classification of each ASV based on a BLAST search is shown, followed by a number in parentheses corresponding to the mean relative abundance of this ASV in infant nasopharyngeal samples (ASV1 was the most abundant ASV across all samples). **c** Mixed effect logistic regression was used to identify URT microbiota features from the preceding sample that predicted pathobiont acquisition. The coefficients from these models, which correspond to the relative effect sizes of associations, are shown for significant associations (*q* < 0.20). ASVs for which higher/lower relative abundances were associated with lower odds of pathobiont acquisition are shown as blue/green bars. Source data are provided as a Source Data file. (AUC-ROC, area under the receiving operating characteristic curve; URT, upper respiratory tract; ASV, amplicon sequence variant).

observed with decreasing age among infants in Indonesia[51]. However, a recent study conducted in the United States reported lower densities among children 6 months of age or younger compared to children 7–24 months of age[52]. Additionally, we found that receipt of PCV-13 was not associated with the colonization density among infants with pneumococcal carriage. O'Brien and colleagues reported that vaccination was associated with a lower colonization density for only vaccine serotypes[53], while the majority of pneumococci that we previously isolated from infants in this cohort were non-vaccine serotypes[54]. However, other studies have not identified an association between vaccination status and pneumococcal colonization density, while some have even reported higher densities among vaccine recipients[55–57]. Finally, we found that URT colonization by *S. aureus* among infants in Botswana is highly seasonal, with acquisition being much higher during the warm months of the rainy season, consistent with seasonal patterns of *S. aureus* skin and soft tissue infections reported in previous pediatric cohorts[58–60].

With increasing recognition of the role the URT microbiota in modifying the risk and severity of ARIs, targeted modification of the URT microbiota is emerging as a promising strategy for ARI prevention. This study demonstrated that respiratory virus infection is associated with shifts in URT microbiota composition and the subsequent acquisition of bacterial pathobionts. More specifically, respiratory virus infection is associated with the acquisition of *H. influenzae*, *M. catarrhalis*, and *S. pneumoniae*, with the acquisition of one of these pathobionts often increasing the odds of acquiring the other two pathobionts. In addition, respiratory virus infection is associated with losses of commensal microbes that appear to contribute to colonization resistance to bacterial pathobionts in the URT. In particular, we found that ASVs classified as *C. propinquum/ pseudodiphtheriticum* (ASV4), *C. accolens/macginleyi* (ASV6), and *C.*

*tuberculostearicum* (ASV10) declined in abundance in association with respiratory virus infection and were associated with lower odds of acquiring *M. catarrhalis*, *S. aureus*, or *S. pneumoniae*. Among healthy Dutch infants, de Steenhuijsen Piters et al. found that early-life respiratory virus infections initiated an interferon-mediated pro-inflammatory response in the URT that was linked to an earlier transition from a microbiota type dominated by *Corynebacterium* and *Dolosigranulum* to microbiota types enriched with *Haemophilus* or *Moraxella*, which in turn were associated with a higher risk of ARIs later in infancy[61]. Our findings are also consistent with those from prior studies suggesting *Corynebacterium* species play a central role in excluding pathobionts from the URT[16,62–64]. For example, Hardy and colleagues reported that *C. pseudodiphtheriticum* strains demonstrate contact-independent killing of *S. aureus* in vitro[64], while Kiryukhina and colleagues found that nasal inoculation of a *C. pseudodiphtheriticum* strain eradicated or reduced the burden of *S. aureus* carriage in healthy volunteers[65]. also found that several streptococcal ASVs were associated with lower odds of pathobiont acquisition, including an ASV classified as *S. thermophilus/vestibularis/salivarius* (ASV15) that declined in abundance with respiratory virus infection and an ASV classified as *Streptococcus mitis/oralis* (ASV9) that was negatively associated with the acquisition of *H. influenzae*, *M. catarrhalis*, and *S. pneumoniae*. Interestingly, antibody responses to *S. mitis* are known to cross-react with *S. pneumoniae*, and intranasal administration of *S. mitis* resulted in lower bacterial loads in a murine model of pneumococcal pneumonia[66]. Moreover, a recent study reported in vitro growth inhibition of nontypeable *H. influenzae* strains by a *S. mitis* strain, although nasal inoculation of this strain did not protect mice from *H. influenzae* pneumonia[67]. We also found that higher abundances of an ASV classified as *D. pigrum* (ASV2) were negatively associated with *S. aureus* acquisition,

consistent with findings from several epidemiological studies and in vitro experiments demonstrating inhibition of *S. aureus* growth by *D. pigrum* strains[17,68–70]. Finally, our analyses suggest a potential antagonistic relationship between *Veillonella massiliensis* and *S. pneumoniae* that warrants further investigation.

We found that the acquisition of *H. influenzae*, *M. catarrhalis*, and *S. pneumoniae* were associated with ARI symptoms independent of respiratory virus detection, with a higher carriage density being additionally associated with ARI symptoms among infants with pneumococcal colonization. URT carriage of these bacterial pathobionts has historically been viewed as an asymptomatic state, although several recent studies have begun to question this traditional dogma. Most notably, experimental human pneumococcal carriage is associated with a robust nasal inflammatory response and symptoms in many participants in the absence of secondary infection[71]. For example, among 96 healthy adults inoculated intranasally with a *S. pneumoniae* serotype 3 strain, more than half (53%) of the 38 colonized participants developed mild symptoms, while symptoms were reported by only 16% of noncolonized participants[72]. Additionally, periods of *H. influenzae*, *M. catarrhalis*, or *S. pneumoniae* colonization were associated with an increase in daily symptoms among adults with chronic obstructive pulmonary disease in the absence of clinical exacerbation[73]. The symptoms associated with the acquisition of bacterial respiratory pathobionts could also be more pronounced in young children than in adults, given that young age is associated with the upregulation of innate and adaptive immune pathways in the URT mucosa of healthy individuals[74,75].

This study has several limitations. First, it was conducted among infants at urban and rural sites in southern Botswana, and our findings may not be generalizable to other age groups or infants in other geographical areas. Additionally, infants were seen for scheduled visits every 1–2 months, and additional visits were not conducted for ARIs between these study visits. Thus, it is likely that some respiratory virus infections occurring among these infants were not identified. Furthermore, some respiratory viruses, such as rhinoviruses and adenoviruses, commonly exhibit prolonged shedding after infection[76,77]; therefore, their detection during study visits may not have indicated an active infection. Most respiratory virus infections were caused by rhinoviruses/enteroviruses, and we lacked the statistical power to evaluate associations between other specific respiratory viruses and bacterial pathobiont colonization. Data from a quantitative PCR assay were only available for *S. pneumoniae*, precluding analyses of the impact of respiratory virus infection on the URT colonization density of the other bacterial pathobionts. Additionally, comparative analyses of URT microbiota composition were based on sample relative abundances, and we are thus unable to determine if the observed associations represent differences in the absolute amounts of bacterial taxa within the URT across samples. This study employed amplicon sequencing of the V4 region of the 16S rRNA gene to characterize the URT bacterial microbiota, which precluded the classification of most sequencing reads to the species level and did not provide data for other microbial kingdoms. Finally, we cannot exclude the possibility of confounding by unmeasured subject characteristics or environmental exposures. Although conditional models would have mitigated confounding by time-invariant variables, we chose to fit mixed-effect models because the strict matching requirements of conditional models would have substantially reduced the effective sample size and introduced a risk of selection bias.

In summary, this study provides important insights into factors that influence URT colonization by bacterial pathobionts that are major causes of ARIs across the lifespan. Our findings contribute to knowledge of microbial ecology in the human URT and demonstrate the influence of viral-bacterial and bacterial-bacterial relationships on the colonization dynamics of these bacterial pathobionts during infancy. An improved understanding of these interspecies relationships could inform the development of novel biotherapeutic strategies that could reduce the global burden of ARIs among both children and adults.

## Methods

### Inclusion and ethics

This study was approved by the Botswana Health Research & Development Committee (HPDME 13/18/1), the Princess Marina Hospital ethics committee (PMH 5/79), and institutional review boards at the University of Pennsylvania (#822692), Duke University Health System (Pro00067434), and McMaster University (#1202). Written informed consent for use of the data and samples as described in this study was obtained from all participants or their legal guardians after a description of the study procedures in their native languages. This study was designed in collaboration with local investigators who were involved throughout the research process and contributed as authors of the resulting publications. Study reporting adheres to the STORMS (STrengthening the Organization and Reporting of Microbiome Studies) guidelines (Supplementary Data 1).

### Study cohort

This prospective cohort study was conducted in Botswana's capital and largest city, Gaborone, which is located in the country's South-East district and has a population of 246,325 based on a census conducted in 2022[78]. Mother-infant dyads ($n = 300$) were enrolled within 72 h of delivery between February 2016 and January 2020 at four sites in Botswana: a referral hospital in Gaborone, a public clinic in a low-income urban neighborhood in Gaborone, a public clinic in a semi-urban area on the outskirts of the city, and a public clinic in a rural village located ~15 km outside of Gaborone. Exclusion criteria included maternal age <18 years, infant birth weight <2000 g, multiple gestation pregnancy, and caesarian delivery. Infants were seen for monthly study visits during the first 6 months of life and every other month thereafter until the infants were 12 months of age. At each visit, participants received compensation (30 Botswana Pula, approximately 3 United States dollars) to offset travel-related expenses. Infants received routine medical care through the public health system, including PCV-13 in a 3 + 0 schedule with doses administered at 2, 3, and 4 months of age. Infants were seen for monthly study visits during the first 6 months of life and every other month thereafter until the infants were 12 months of age. At all visits, nasopharyngeal swab samples were collected from mothers and infants and a caregiver questionnaire was administered to evaluate for ARI symptoms, defined by the presence of cough, rhinorrhea, or nasal congestion. Nasopharyngeal samples were placed directly into MSwab medium (Copan Italia), transported to the National Health Laboratory in Gaborone, divided into aliquots, and frozen within 4 h of collection to -80 °C. Nasopharyngeal sample aliquots were shipped in batches on dry ice to the designated research laboratories for this study. Testing of infant nasopharyngeal samples for respiratory viruses was performed using real-time PCR assays, as previously described[2]. Testing for bacterial pathobionts (*H. influenzae*, *M. catarrhalis*, *S. aureus*, and *S. pneumoniae*) was similarly performed using a custom multiplex real-time PCR assay developed by DB (Supplementary Table 4). Pneumococcal colonization density was estimated using a quantitative PCR assay targeting the autolysin (*lytA*) gene[62]. All PCR assays were performed by the Infectious Disease Research Group in the Research Institute of St. Joe's Hamilton (Hamilton, Ontario, Canada).

### Processing of samples for 16S ribosomal RNA gene sequencing

Infant nasopharyngeal samples ($n = 2409$) underwent amplicon sequencing of the V4 region of the 16S rRNA gene in eight batches at Duke University (Durham, North Carolina, United States). DNA was

extracted using Powersoil Pro extraction kits (Qiagen). DNA concentrations were determined using Qubit dsDNA high-sensitivity assay kits (Thermo Fisher Scientific). Negative extraction and PCR controls were amplified with all eight batches of samples included in analyses to evaluate for background contamination. For the first two sample batches, these negative controls were verified to not have visible bands on gel electrophoresis. For the remaining six batches, sequencing was performed on negative control samples. PCR amplification of the V4 region of the 16S rRNA gene was performed using primers 515 F and 806 R, and amplicon pools were sequenced using a MiSeq instrument (Illumina) configured for 2 × 250 bp sequencing.

### Bioinformatic processing of 16S rRNA sequencing data

Paired-end reads were filtered and trimmed (maxEE = 2; truncLen = 175/125 bp), merged, denoised, chimera filtered, and binned into ASVs using DADA2 v1.16.0[79]. ASVs were given provisional taxonomic assignments using the DADA2 implementation of the naïve Bayesian classifier (minBoot = 80) and based on alignment to the expanded Human Oral Microbiome Database v15.1[80]. Negative control samples had a median (IQR) of 500 (293–961) reads, with the most abundant ASVs classified as common reagent contaminants, such as *Escherichia coli* (ASV47, 30.2% of control sample reads), *Pelomonas aquatica* (ASV76, 11.3%), and *Bradyrhizobium* sp. (ASV313, 5.6%)[81–83]. Identification and removal of potential reagent contaminants and batch variability correction were performed using a two-tier strategy. First, contaminant ASVs were identified and removed using two approaches provided by the *decontam* R package v1.24.0: one based on detection in negative control samples (prevalence method) and another based on a negative correlation between the detection frequency and the input library DNA concentration (frequency method, threshold = 0.10)[84]. Second, contaminant ASVs were identified and removed based on differential prevalence across the eight sequencing batches ($k = 0.05$) as described by Moossavia and colleagues[81]. Samples with <1000 sequencing reads after contaminant removal were excluded from further analyses. Sequencing reads were classified into 18,773 ASVs representing 248 genera from 12 phyla. ASVs were assigned numbers in ascending order based on mean relative abundance (ASV1 was the most abundant ASV across all samples in the final dataset). PICRUSt2 v2.5.2[26] was used to predict the abundances of metabolic pathways and enzymes in the MetaCyc database v27.1[85].

### Statistical models for analyses of molecular data on respiratory viruses and bacterial pathobionts

Logistic regression was used to evaluate associations between infant age and the prevalence of bacterial pathbiont colonization as detected by PCR. To evaluate associations between the detection of respiratory viruses or bacterial pathobionts and the presence of ARI symptoms, a mixed effect logistic regression model was fit with subject as a random effect to account for repeated sampling of individuals. Mixed effect logistic regression was similarly used to evaluate associations between respiratory virus infection, preceding pathobiont colonization, and infant and sample characteristics and the acquisition of specific bacterial pathobionts. These models included subject as a random effect and the following infant and sample characteristics as fixed effects: age, sex, low birth weight (<2500 g), maternal HIV infection, location of residence (urban vs. rural), number of children under 5 years of age in the household, season (time-varying; dry vs. rainy), breastfeeding (time-varying), receipt of antibiotics since the prior study visit (time-varying), and PCV-13 doses (time-varying; for *S. pneumoniae* only). Linear regression adjusting for these same variables was used to evaluate the impact of respiratory virus infection, pathobiont colonization, and infant and sample characteristics on the pneumococcal colonization density.

### Statistical methods for analyzing infant upper respiratory microbiota composition

K-medoids clustering on Bray-Curtis distances was used to group samples into distinct microbiota types, with the optimal number of microbiota types chosen using the Calinski-Harabasz index. To assess the robustness of the eight identified microbiota types to clustering algorithm and distance metric, samples were also grouped into eight microbiota types using k-means clustering and the default squared Euclidean distance metric. Mixed effect logistic regression was used to evaluate associations between infant age and the prevalence of each microbiota type. Similarly, mixed effect logistic regression models were fit to evaluate associations between microbiota type and the acquisition of specific bacterial pathobionts with subject as a random effect and adjusting for infant and sample characteristics as fixed effects. Analyses of URT microbiota composition conducted at the ASV level were limited to 31 ASVs present at a relative abundance at or above 0.1% in at least 5% of samples. To assign species information to each of these ASVs, we performed a standard nucleotide REFSEQ BLAST search using the National Center for Biotechnology Information's Bacteria and Archaea 16S ribosomal RNA project database[86] and used a best-hit approach based on the *E*-value with a minimum percent identity of 97%. We used MaAsLin2 v1.18.0[87] to fit linear mixed effect models evaluating associations between respiratory virus detection by PCR and the centered log ratio-transformed relative abundances of ASVs within the URT microbiota. Models included subject as a random effect and the following fixed effects: age (cubic b-spline of the age in months), sex, low birth weight, maternal HIV infection, location of residence, the number of children under 5 years of age in the household, season (time-varying; dry vs. rainy), breastfeeding (time-varying), receipt of antibiotics since the prior study visit (time-varying), and PCV-13 doses (time-varying; for *S. pneumoniae* only). We evaluated the utility of the composition of the preceding URT microbiota in predicting the acquisition of bacterial pathobionts at the current study visit by applying the random forest algorithm with *randomForest* R package v4.7.1.1, restricting analyses to preceding samples testing negative for the pathobiont of interest by PCR. Prediction accuracy was evaluated using AUC-ROC with leave-one-subject-out cross-validation. Total sum scaling was used to normalize ASV count data to relative abundances before natural log transformation, with zeros replaced by 0.5 before conversion to relative abundances. For analyses of microbial pathway relative abundances, zeros were replaced by values corresponding to half of the sample-specific minimum. To address the unequal distribution of events of pathobiont acquisition, we configured the random forests to use stratified sampling. The reported feature importance represents the average calculated across all leave-one-subject-out cross-validation runs. To identify specific ASVs that offered the highest marginal association with bacterial pathobiont acquisition while adjusting for infant age, we employed mixed effect logistic regression with the log relative abundance of each ASV and the cubic b-spline of age in months as predictors, pathobiont acquisition as the outcome, and a random effect to account for repeated measurements from the same subject. For all analyses, listwise deletion was used to exclude samples or time intervals with missing data for any variable. The Benjamini-Hochberg procedure was used to control the false discovery rate for analyses involving multiple comparisons ($q < 0.20$). All analyses were conducted using R v4.4.2.

### Reporting summary

Further information on research design is available in the Nature Portfolio Reporting Summary linked to this article.

## Data availability

All data needed to reproduce the findings of this study are publicly available. The 16S rRNA gene sequencing data generated in this study

have been deposited in the Sequence Read Archive under accession number PRJNA698366. All metadata and processed files supporting analyses are available on GitHub [https://github.com/mskelly7/Pathobiont_colonization][88]. All files used to generate the figures in the main manuscript and Supplementary Information are included in the Source Data file. Source data are provided with this paper.

## Code availability

The scripts used to process the raw 16S rRNA gene sequencing table, perform all data analyses, and generate the tables and figures presented in this manuscript and in the Supplementary Information are available on GitHub [https://github.com/mskelly7/Pathobiont_colonization][88].

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

## Acknowledgements
We offer sincere gratitude to the individuals who participated in this research. We would like to thank Copan Italia (Brescia, Italy) for the donation of the MSwab media and flocked swabs used in collecting nasopharyngeal specimens. We also thank the Duke Microbiome Core Facility for performing the DNA extractions and library preparations, as well as the Duke Sequencing and Genome Technologies Core Facility for sequencing these libraries. This research was supported in part by a research grant (#60201, M.S.K.) from Investigator-Initiated Studies Program of Merck & Co., Inc. The opinions expressed in this paper are those of the authors and do not necessarily represent those of Merck & Co., Inc. Additional support for this work was provided by a grant from the Burroughs Wellcome Fund and the American Society of Tropical Medicine and Hygiene (M.S.K.), Children's Hospital of Philadelphia and the Pincus Family Foundation, and through core services from the Penn Center for AIDS Research, a National Institutes of Health (NIH)-funded program (P30-AI045008). M.S.K. was supported by a NIH Career Development Award (K23-AI135090) and a research grant from the Society for Pediatric Research.

## Author contributions
Conceptualization: M.S.K., P.C.S., M.S. Methodology: M.S.K., P.S., A.P.S., C.K.C., J.F.R., J.H.H., P.C.S., D.B., M.S. Investigation: M.S.K., S.C.B., E.Q., T.M., M.Z.P., K.L., J.G., J.M. Visualization: M.S.K., P.S. Funding acquisition: M.S.K., A.P.S. Project administration: M.S.K. Supervision: M.S.K. Writing—original draft: M.S.K. Writing—review and editing: P.S., S.C.B., E.Q., A.P.S., T.M., M.Z.P., C.K.C., J.F.R., K.L., J.G., J.M., J.H.H., P.C.S., D.B., M.S.

## Competing interests
M.S.K. is a consultant for Merck & Co, Inc. and Invivyd. All other authors declare that they have no competing interests.
