## [Peer Review file · Nature Communications]

Role of the upper respiratory microbiota in respiratory virus and bacterial pathobiont dynamics

Corresponding Author: Dr Matthew Kelly

Version 0:

Reviewer comments:

Reviewer #1

(Remarks to the Author)

There are some interesting findings here regarding pathogen interactions which no doubt be of interest to a clinical audience. Some general comments regarding the statistical methodology.

1. The results section contains a very large number of p-values. The authors must be aware of the multiple testing issue. Perhaps just present effect sizes and confidence intervals. Given the multitude of comparisons I would regard this study as exploratory in nature and could be useful in generating clear hypotheses to be tested subsequently.

2. When looking at pathogen interactions, it is difficult to distinguish between interactions at the host level, i.e. where individuals are generally more or less susceptible to infection due to environment etc and interactions at the pathogen level i.e. where the presence of one pathogen directly influences the risk of acquisition of another. This latter interaction is of more interest and is assumed by the authors to explain the interaction effects observed.

Longitudinal data potentially enables the two types of interaction to be distinguished...however the investigators' use of mixed effect logistic regression does not wholly exclude the possibility that the observed effects are not partially host-level interactions. Although the models controlled for fixed host characteristics, there is always the possibility of unmeasured confounding. I would prefer the use of within-person models, specifically conditional logistic regression, where all between-person time invariant confounding is automatically eliminated.

One minor point.....the authors should not say "no association" when the observed association fell short of an arbitrary level of statistical significance.

(Remarks on code availability)

Reviewer #2

(Remarks to the Author)

The authors have produced a well written manuscript on a valuable and interesting topic. The study is well conducted and involves a large sample of children in Botswana with extensive longitudinal follow-up across the first year of life. The methods and statistical analyses are clearly described. The results show cooperative relationships between the pathobionts *H. influenzae*, *M. catarrhalis* and *S. pneumoniae* in the human upper respiratory tract (URT) and also describe in detail the relationship between pathobiont acquisition and shifts in microbiota composition with respiratory virus presence in early life. In particular, consistent with previous work in the field, the authors findings suggest *Corynebacterium* species have a protective role in reducing pathobiont acquisition in the upper respiratory tract. With the increasing recognition of the important role the URT plays in modifying respiratory tract infections the results of this study add beneficial and important information to the growing body of research in this field.

However, I do note the study is not unique in the conducting of longitudinal follow-up of the URT viral and microbiome relationship in infants over the first year of life. For instance the Microbiome Utrecht Infant Study has also demonstrated similar changes in *Corynebacterium* detection with early life respiratory viral infections (REFS: de Steenhuijsen Piters, et al.

Early-life viral infections are associated with disadvantageous immune and microbiota profiles and recurrent respiratory infections. Nat Microbiol 2022; 7:224–237 and Bosch AATM et al. Maturation of the Infant Respiratory Microbiota, Environmental Drivers, and Health Consequences. A Prospective Cohort Study. Am J Respir Crit Care Med. 2017 Dec 15;196(12):1582-1590.) As far as I can tell the authors have not included reference to this work in their manuscript. The authors may wish to consider including reference to the results from the Utrecht cohort in the discussion of their findings as it may support the relevance of the work undertaken in Botswana in comparison to a European infant cohort.

My only other minor comment in relation to the manuscript is regarding analysis of the impact of virus detection on bacterial pathobiont presence. For supplementary tables 1 and 2 I note that for some viruses (e.g. adenovirus and rhino/enterovirus) the detection of a virus was more likely not to be associated with ARI symptoms than to be associated with ARI symptoms at the time of detection. Given that viral detection on PCR from nasopharyngeal samples can persist for weeks following acute infection it is possible that virus detection does not always represent a concurrent viral respiratory tract infection but could indicate a previous infection. Did the authors perform an analysis of correlation for symptomatic and non-symptomatic virus detection and bacterial pathobionts detection separately? I appreciate numbers may be too small to draw any inferences for this type of analysis but it would be interesting to determine whether bacterial pathobiont detection is more associated with symptomatic viral detection or with non-symptomatic viral detection (presumed recovering post viral infection) as this could have relevance for proposed interventions going forwards.

(Remarks on code availability)

All data files are listed appropriately however I have not been able to run any of the .RDS files. This may be due to an issue with my computer.

Reviewer #3

(Remarks to the Author)

In this study, Kelly et al., used 2,409 nasopharyngeal swabs from 300 infants to investigate relationships between respiratory viruses, the URT bacterial microbiota, and bacterial respiratory pathobionts. Nasopharyngeal swab samples were collected from infants in Botswana who were followed during the first year of life. PCR was used for respiratory viruses and bacterial pathogens. Then 16S rRNA gene sequencing was performed on isolated DNA

This study tries to deal with the gap that few studies investigated microbe-microbe interactions particularly affecting bacterial pathobionts using a longitudinal study design and culture independent methods. The data on the potential collaborative relationship between respiratory pathogens and viruses is quite interesting.

Below are my point-by-point comments.

Major comments:

1. Regarding removal of contaminants: “contaminant ASVs were identified and removed based on the presence in negative control samples or negative correlation with DNA concentration using the frequency method”. What was used to measure concentration of bacterial DNA? Total DNA would not be a good measurement of microbial DNA in these nasopharyngeal swabs.
2. Where is the data about the negative controls?
3. How were the nasopharyngeal swabs collected, stored, transported?
4. Line 103: “Infants were born vaginally”. All of them?
5. Line 198: “In order to investigate shifts in overall URT microbiota community structure, we used k-medoids clustering on Bray-Curtis distances to classify infant nasopharyngeal samples into eight distinct microbiota types (Figure 3a,b)”. I assume this is based on the 16S rRNA gene sequencing data. Please clarify.
6. Line 203: “Excluding samples collected during the birth visit and with subject included as a random effect, the prevalence of microbiota types dominated by Staphylococcus (mixed effect...”. How was “dominance” defined, what threshold were used. Were those the same among each taxa mentioned including those involving more than one taxa such as Corynebacterium/Dolosigranulum/Moraxella.
7. Were there differences in how dominant each of those taxa tended to be in the 16S data?
8. How well the targeted PCR data on pathobionts correlated with the relative abundance of those taxa in the 16S data?
9. There is no data on the paired mother samples. Were those available? If they were I would have expected several analyses related to carriage and cross pollination of microbes between mother and child.
10. Several of the children received antibiotics. How did those antibiotic exposures change the 16S rRNA gene sequencing data (might have to be limited to some antibiotic class).

Minor Comments

1. Line 190: “lack of associations between *S. aureus* and other pathobionts in longitudinal analyses suggests that the negative co-occurrence patterns observed between *S. aureus* and other bacterial pathobionts are unlikely to reflect direct antagonistic interspecies interactions”. Not sure that I follow the rationale.

(Remarks on code availability)

Version 1:

Reviewer comments:

Reviewer #1

(Remarks to the Author)

Thankyou for the revision. All my issues have been addressed.

(Remarks on code availability)

Reviewer #2

(Remarks to the Author)

The authors have appropriately responded to my previous comments with inclusion of additional references and analyses. I am satisfied that the additional analysis evaluating associations between symptomatic and asymptomatic respiratory virus infections is appropriate and will be of interest to the reader.

I have no other additional comments to make.

(Remarks on code availability)

Reviewer #3

(Remarks to the Author)

The authors have addressed my comments

(Remarks on code availability)

I scanned through and it seems that it appropriate

Response to Reviewers

Reviewer #1 (Remarks to the Author):

There are some interesting findings here regarding pathogen interactions which no doubt be of interest to a clinical audience. Some general comments regarding the statistical methodology.

1. The results section contains a very large number of p-values. The authors must be aware of the multiple testing issue. Perhaps just present effect sizes and confidence intervals. Given the multitude of comparisons I would regard this study as exploratory in nature and could be useful in generating clear hypotheses to be tested subsequently.

Throughout the manuscript, we have removed p-values where odds ratios and confidence intervals are presented. However, we have retained p-values in cases where effect sizes and confidence intervals are not presented because they are not easily interpretable.

2. When looking at pathogen interactions, it is difficult to distinguish between interactions at the host level, i.e. where individuals are generally more or less susceptible to infection due to environment etc and interactions at the pathogen level i.e. where the presence of one pathogen directly influences the risk of acquisition of another. This latter interaction is of more interest and is assumed by the authors to explain the interaction effects observed.

Longitudinal data potentially enables the two types of interaction to be distinguished....however the investigators' use of mixed effect logistic regression does not wholly exclude the possibility that the observed effects are not partially host-level interactions. Although the models controlled for fixed host characteristics, there is always the possibility of unmeasured confounding. I would prefer the use of within-person models, specifically conditional logistic regression, where all between-person time invariant confounding is automatically eliminated.

We appreciate the reviewer's insightful comment and agree conditional models offer the advantage of not being prone to confounding by unmeasured host-level variables. However, after thoroughly evaluating their application for analyses of these study data, we continue to favor the use of (logistic and linear) mixed-effect models for several reasons.

First, conditional logistic regression excludes any stratum (subject) that lacks outcome variation. For instance, in analyses of factors associated with pathobiont acquisition, conditional logistic regression omits data from infants who do not acquire the pathobiont during the study period or who acquire the pathobiont during the first month of life and remain colonized throughout follow-up. Given the frequency of these scenarios in bacterial pathobiont dynamics during infancy, such exclusions substantially reduce the effective sample size and may introduce bias if the excluded strata systematically differ from those retained.

Second, we found that some conditional models failed to converge due to variables perfectly predicting the outcome within certain strata (subjects), often referred to as complete separation. The most common way to address this issue is to remove these strata, which would further decrease the effective sample size and increase the risk of bias.

Third, while conditional models eliminate between-subject confounding by time-invariant factors, they also prevent estimation of these effects. This limitation is particularly relevant to our analysis, as it would preclude us from assessing associations between key host characteristics and environmental exposures—such as maternal HIV status, presence of young children, and exposure to smoke from solid fuels—and pathobiont acquisition during infancy.

For these reasons, we continue to use (logistic and linear) mixed-effect models in the revised manuscript. We believe that accounting for subject-level clustering is essential for analyzing this dataset. However, our evaluation of conditional models revealed that the reduced effective sample size, potential bias introduced

by strict matching, and the inability to estimate time-invariant effects limit our ability to draw meaningful conclusions. Nonetheless, we acknowledge the reviewer's concern regarding potential unmeasured confounding and have added the following sentence to the Discussion section:

Page 19: *“Finally, we cannot exclude the possibility of confounding by unmeasured subject characteristics or environmental exposures. Although conditional models would have mitigated confounding by time-invariant variables, we chose to fit mixed-effect models because the strict matching requirements of conditional models would have substantially reduced the effective sample size and introduced a risk of selection bias.”*

One minor point.....the authors should not say “no association” when the observed association fell short of an arbitrary level of statistical significance.

We thank the reviewer for these important point and agree that these statements do not reflect what is an arbitrary level of statistical significance. We have modified the following sentences to address this concern:

Page 8: *“The odds of acquiring *S. pneumoniae* were higher in urban settings compared to rural settings (OR: 1.56, 95% CI: 1.09–2.23) but were not significantly associated with the number of 13-valent pneumococcal conjugate vaccine (PCV-13) doses received (OR: 1.15, 95% CI: 0.91–1.44).”*

Page 8: *“PCV-13 vaccination status was not significantly associated with the density of pneumococcal colonization [median (IQR) log₁₀ copies/mL; 0 doses: 7.5 (6.7-7.9); 1 dose: 7.3 (6.6-7.6); 2 doses: 7.1 (6.1-7.7); 3 doses: 7.0 (6.0-7.7); p=0.14].”*

Page 16: *“However, other studies have not identified an association between vaccination status and pneumococcal colonization density, while some have even reported higher densities among vaccine recipients.”*

Reviewer #2 (Remarks to the Author):

The authors have produced a well written manuscript on a valuable and interesting topic. The study is well conducted and involves a large sample of children in Botswana with extensive longitudinal follow-up across the first year of life. The methods and statistical analyses are clearly described. The results show cooperative relationships between the pathobionts *H. influenza*, *M. catarrhalis* and *S. pneumoniae* in the human upper respiratory tract (URT) and also describe in detail the relationship between pathobiont acquisition and shifts in microbiota composition with respiratory virus presence in early life. In particular, consistent with previous work in the field, the authors findings suggest *Corynebacterium* species have a protective role in reducing pathobiont acquisition in the upper respiratory tract. With the increasing recognition of the important role the URT plays in modifying respiratory tract infections the results of this study add beneficial and important information to the growing body of research in this field.

We sincerely appreciate the reviewer's thoughtful and positive feedback on this manuscript.

However, I do note the study is not unique in the conducting of longitudinal follow-up of the URT viral and microbiome relationship in infants over the first year of life. For instance the Microbiome Utrecht Infant Study has also demonstrated similar changes in *Corynebacterium* detection with early life respiratory viral infections (REFS: de Steenhuijsen PETERS, et al. Early-life viral infections are associated with disadvantageous immune and microbiota profiles and recurrent respiratory infections. *Nat Microbiol* 2022; 7:224–237 and Bosch AATM et al. Maturation of the Infant Respiratory Microbiota, Environmental Drivers, and Health Consequences. *A Prospective Cohort Study. Am J Respir Crit Care Med.* 2017 Dec 15;196(12):1582-1590.) As far as I can tell the authors have not included reference to this work in their manuscript. The authors may wish to consider including reference to the results from the Utrecht cohort in the discussion of their findings as it may

support the relevance of the work undertaken in Botswana in comparison to a European infant cohort.

We thank the reviewer for highlighting the important work that has been published by this group, and we agree that their findings are relevant to the current analysis. In particular, we now discuss a study by de Steenhuijsen Piters et al. which used a multi-omic approach to show that early-life respiratory virus infections trigger changes in host mucosal immunity that are linked to URT microbiota shifts similar to those we observed in our cohort.

Page 17: *“Among healthy Dutch infants, de Steenhuijsen Piters et al. found that early-life respiratory virus infections initiated an interferon-mediated pro-inflammatory response in the URT that was linked to an earlier transition from a microbiota type dominated by Corynebacterium and Dolosigranulum to microbiota types enriched with Haemophilus or Moraxella, which in turn were associated with a higher risk of ARIs later in infancy.”*

My only other minor comment in relation to the manuscript is regarding analysis of the impact of virus detection on bacterial pathobiont presence. For supplementary tables 1 and 2 I note that for some viruses (e.g. adenovirus and rhino/enterovirus) the detection of a virus was more likely not to be associated with ARI symptoms than to be associated with ARI symptoms at the time of detection. Given that viral detection on PCR from nasopharyngeal samples can persist for weeks following acute infection it is possible that virus detection does not always represent a concurrent viral respiratory tract infection but could indicate a previous infection.

This is an excellent point. It is well recognized that certain respiratory viruses, such as rhinoviruses and adenoviruses, can exhibit prolonged shedding, often persisting for weeks or even months. As the reviewer has noted, detecting these viruses during study visits does not necessarily indicate an active infection. To ensure that this is conveyed to readers, we have added the following sentence to the paragraph discussing the study's limitations:

Page 19: *“Furthermore, some respiratory viruses, such as rhinoviruses and adenoviruses, commonly exhibit prolonged shedding after infection; therefore, their detection during study visits may not have indicated an active infection.”*

Did the authors perform an analysis of correlation for symptomatic and non-symptomatic virus detection and bacterial pathobionts detection separately? I appreciate numbers may be too small to draw any inferences for this type of analysis but it would be interesting to determine whether bacterial pathobiont detection is more associated with symptomatic viral detection or with non-symptomatic viral detection (presumed recovering post viral infection) as this could have relevance for proposed interventions going forwards.

We thank the reviewer for this insightful comment. We now include analyses evaluating associations between symptomatic (accompanied by ARI symptoms) and asymptomatic respiratory virus infections and both the odds of acquiring bacterial pathobionts and the pneumococcal colonization density. We present results of these new analyses in Supplemental Table 2 and in the following sentences in the Results section of the revised manuscript:

Page 7: *“The odds of acquiring H. influenzae, M. catarrhalis, and S. pneumoniae increased in association with respiratory virus detections accompanied by ARI symptoms (n=287) but not with asymptomatic respiratory virus detections (n=352) (Supplementary Table 2).”*

Page 7: *“The density of pneumococcal colonization tended to be higher among infants with symptomatic compared to asymptomatic respiratory virus detections [median (IQR): 7.5 (6.9-7.9) vs. 7.3 (6.5-7.8) log₁₀ copies/mL; p=0.07].”*

Additionally, we have added the following discussion of these results to the Discussion section of the revised manuscript:

*Pages 14-15: “Notably, we found that respiratory virus detections accompanied by ARI symptoms were associated with increased odds of acquiring *H. influenzae*, *M. catarrhalis*, and *S. pneumoniae*, whereas this association was not observed with asymptomatic respiratory virus detections. Moreover, among infants with *S. pneumoniae* carriage, colonization density tended to be higher in those with symptomatic respiratory virus detections compared to those with asymptomatic respiratory virus detections. These findings are consistent with prior studies suggesting that secondary bacterial infections are more likely to occur following symptomatic respiratory virus infections. For instance, among 367 infants followed during the first year of life, symptomatic respiratory virus infections were associated with an increased risk of acute otitis media, whereas this association was not observed for asymptomatic respiratory virus infections. Symptoms of respiratory virus infections typically result from loss of cellular tight junctions, vascular leakage and edema, increased mucus production, and sloughing of epithelial cells, all of which can promote bacterial pathobiont adhesion and overgrowth.”*

Reviewer #2 (Remarks on code availability):

All data files are listed appropriately however I have not been able to run any of the .RDS files. This may be due to an issue with my computer.

We have re-uploaded all files to the study’s GitHub repository, and verified that the metadata, processed sequencing data, and analysis scripts are accurate and successfully reproduce the results and figures presented in the revised manuscript. However, we are happy to help the reviewers access or run these files if the issues persist.

Reviewer #3 (Remarks to the Author):

In this study, Kelly et al., used 2,409 nasopharyngeal swabs from 300 infants to investigate relationships between respiratory viruses, the URT bacterial microbiota, and bacterial respiratory pathobionts. Nasopharyngeal swab samples were collected from infants in Botswana who were followed during the first year of life. PCR was used for respiratory viruses and bacterial pathogens. Then 16S rRNA gene sequencing was performed on isolated DNA

This study tries to deal with the gap that few studies investigated microbe-microbe interactions particularly affecting bacterial pathobionts using a longitudinal study design and culture independent methods. The data on the potential collaborative relationship between respiratory pathogens and viruses is quite interesting.

We appreciate the reviewer’s supportive comments.

Below are my point-by-point comments.

Major comments:

1. Regarding removal of contaminants: “contaminant ASVs were identified and removed based on the presence in negative control samples or negative correlation with DNA concentration using the frequency method”. What was used to measure concentration of bacterial DNA? Total DNA would not be a good measurement of microbial DNA in these nasopharyngeal swabs.

We identified reagent contaminants using *decontam*, an open-source R package that was validated in a 2018 *Microbiome* paper and has since been cited in >2,500 publications. For the frequency method, *decontam* uses the total DNA concentrations of libraries, leveraging the fact that in low-biomass samples (i.e., lower total DNA), contaminating microbial DNA will tend to constitute a larger fraction of the total DNA. While we agree that the bacterial DNA concentration would likely correlate better with the frequency of reagent contaminants in 16S rRNA sequencing data, we are unaware of a reliable method to accurately quantify

bacterial DNA in mixed samples; while PCR of the 16S rRNA gene has been used for this purpose, the amplification process introduces biases that could affect the results. To clarify our use of *decontam* in this analysis, we have revised this sentence in the manuscript to the following:

Pages 21-22: “First, contaminant ASVs were identified and removed using two approaches provided by the *decontam R* package v1.24.0: one based on detection in negative control samples (prevalence method) and another based on a negative correlation between the detection frequency and the input library DNA concentration (frequency method, threshold=0.10).”

2. Where is the data about the negative controls?

We have added a sentence in the Methods section that presents the data for negative control samples:

Page 21: “Negative control samples had a median (IQR) of 500 (293-961) reads, with the most abundant ASVs classified as common reagent contaminants, including *Escherichia coli* (ASV47, 30.2% of control sample reads), *Pelomonas aquatica* (ASV76, 11.3%), and *Bradyrhizobium sp.* (ASV313, 5.6%).”

3. How were the nasopharyngeal swabs collected, stored, transported?

We appreciate the reviewer having pointed out this omission. We have added the following sentences to the Methods section of the revised manuscript:

Page 20: “Nasopharyngeal samples were placed directly into MSwab medium (Copan Italia), transported to the National Health Laboratory in Gaborone, divided into aliquots, and frozen within 4 hours of collection to -80°C. Nasopharyngeal sample aliquots were shipped in batches on dry ice to the designated research laboratories for this study.”

4. Line 103: “Infants were born vaginally”. All of them?

Thank you for the chance to clarify this point. We chose to include Caesarian delivery as an exclusion criterion for this study for a few reasons: (1) previous studies have documented the influence of delivery mode on the upper respiratory microbiota, and (2) we aimed to ensure that our findings reflect the majority of infants born in low- and middle-income countries (LMICs). Current data suggest that the C-section rate in LMICs is ~10-15%, compared to a C-section rate exceeding 30-40% in many high-income countries.

5. Line 198: “In order to investigate shifts in overall URT microbiota community structure, we used k-medoids clustering on Bray-Curtis distances to classify infant nasopharyngeal samples into eight distinct microbiota types (Figure 3a,b)”. I assume this is based on the 16S rRNA gene sequencing data. Please clarify.

Thank you for pointing out this omission. The reviewer is correct that this analysis was based on the 16S rRNA gene sequencing data. We have revised this sentence to clarify this point:

Page 10: “To investigate shifts in overall URT microbiota community structure, we analyzed 16S rRNA gene sequencing data and applied k-medoids clustering on Bray-Curtis distances to classify infant nasopharyngeal samples into eight distinct microbiota types (Figure 3a,b).”

6. Line 203: “Excluding samples collected during the birth visit and with subject included as a random effect, the prevalence of microbiota types dominated by Staphylococcus (mixed effect...)”. How was “dominance” defined, what threshold were used. Were those the same among each taxa mentioned including those involving more than one taxa such as Corynebacterium/Dolosigranulum/Moraxella.

To prevent confusion for readers, we no longer use the term dominance. Instead, we specify that URT microbiota types are named based on the most abundant genus or genera.

7. Were there differences in how dominant each of those taxa tended to be in the 16S data?

Yes, the differences in what we previously referred to as dominance were determined by the mean relative abundances of bacterial genera in the URT microbiota types, as identified from the 16S rRNA gene sequencing data.

8. How well the targeted PCR data on pathobionts correlated with the relative abundance of those taxa in the 16S data?

We were also interested in this question and have summarized the findings in Supplemental Figure 3. Higher relative abundances of ASVs 7 and 8 were observed with *H. influenzae* colonization ($p < 0.0001$ for both analyses), ASV1 with *M. catarrhalis* colonization ($p < 0.0001$), ASV3 with *S. aureus* colonization ($p < 0.0001$), and ASV5 with *S. pneumoniae* colonization ($p < 0.0001$), suggesting that these ASVs contain the corresponding pathobionts. Additionally, a strong positive correlation ($\rho = 0.83$, $p < 0.0001$) was observed between the pneumococcal colonization density (determined by quantitative PCR) and the relative abundance of ASV5, further supporting that this ASV contains *S. pneumoniae*. However, many samples in which these pathobionts were detected also had non-zero relative abundances for the corresponding ASV(s), suggesting that these ASVs likely include other species from the same genus.

9. There is no data on the paired mother samples. Were those available? If they were I would have expected several analyses related to carriage and cross pollination of microbes between mother and child.

We conducted 16S rRNA gene sequencing on maternal URT samples collected during the birth visit and have previously published our combined analysis of maternal and infant microbiota data (reference included below). In this current study, our objective was to evaluate the extent to which the URT bacterial microbiota promotes colonization by bacterial respiratory pathobionts following respiratory virus infection.

Kelly MS, Plunkett C, Yu Y, Aquino JN, Patel SM, Hurst JH, et al. Non-diphtheriae *Corynebacterium* species are associated with decreased risk of pneumococcal colonization during infancy. *The ISME Journal*. 2021:1-11.

10. Several of the children received antibiotics. How did those antibiotic exposures change the 16S rRNA gene sequencing data (might have to be limited to some antibiotic class).

We previously analyzed the impact of antibiotics on the nasopharyngeal microbiota during infancy using data from this cohort. In summary, antibiotic exposure was associated with a decrease in the relative abundances of bacterial genera generally linked to respiratory health, such as *Corynebacterium* and *Lactobacillus*, and increases in the relative abundances of genera containing common respiratory pathobionts (*Haemophilus*, *Moraxella*, *Streptococcus*).

Kelly MS, Plunkett C, Yu Y, Aquino JN, Patel SM, Hurst JH, et al. Non-diphtheriae *Corynebacterium* species are associated with decreased risk of pneumococcal colonization during infancy. *The ISME Journal*. 2021:1-11.

Minor Comments

1. Line 190: “lack of associations between *S. aureus* and other pathobionts in longitudinal analyses suggests that the negative co-occurrence patterns observed between *S. aureus* and other bacterial pathobionts are unlikely to reflect direct antagonistic interspecies interactions”. Not sure that I follow the rationale.

We apologize for the confusion and have revised this sentence to more clearly convey our intended point:

Page 9: “Moreover, the absence of associations between *S. aureus* and other bacterial pathobionts in longitudinal analyses suggests that the negative co-occurrence patterns observed between *S. aureus* and these species in cross-sectional analyses of individual samples are unlikely to result from direct antagonistic interspecies interactions.”